# CERES-Maize (DSSAT) Model Applications for Maize Nutrient Management Across Agroecological Zones: A Systematic Review

**DOI:** 10.3390/plants14050661

**Published:** 2025-02-21

**Authors:** Addey Gobezie, Dereje Ademe, Lakesh K. Sharma

**Affiliations:** 1Department of Plant Science, College of Agriculture and Natural Recourse, Debre Maros University, Amhara P.O. Box. 269, Ethiopia; 2Soil, Water and Ecosystem Sciences Department, Institute of Food and Agricultural Systems, University of Florida, Gainesville Campus, Gainesville, FL 32611, USA

**Keywords:** *Zea mays* L., CERES-maize, DSSAT, nutrient management, agroecology

## Abstract

Effective nutrient management is essential for boosting maize yield and quality and tackling factors that limit or reduce productivity. The Crop Environment Resource Synthesis (CERES)-Maize model embedded in the Decision Support Systems for Agrotechnology Transfer (DSSAT) cropping system model (CSM), known for its accurate predictions, serves as a valuable tool for guiding agricultural decisions, particularly in nutrient management, offering an efficient alternative to traditional long term field trials. This systematic review consolidates the current knowledge on nutrient management practices for maize using the CERES-Maize (DSSAT) model, providing insights that benefit researchers, agronomists, policymakers, and farmers. By leveraging crop system, soil carbon and nitrogen, and daily water balance models with crop/land management options, the model accurately predicts the effect of agricultural practices on crop growth, yield, and environmental impacts. This enables the evaluation of diverse management strategies to improve productivity and sustainability.

## 1. Introduction

Maize (*Zea mays* L.) stems from wild grasses in Mexico 7000 years ago and was domesticated as a vital food source by Native Americans [1]. Today, maize stands as one of the world’s most significant crops, shaping societies, economies, and diets [2]. As a globally cultivated cereal grain, maize plays a critical role in food security, particularly in Africa, where it serves as a staple food [3]. It can be processed into various products like flour, snacks, and cornflakes, as well as commercial goods involving starch, sugar, oil, beverages, adhesives, industrial alcohol, and ethanol [4]. Additionally, maize is essential for animal feed, biofuel production, and numerous industrial applications [5]. Its widespread cultivation and versatility make maize indispensable to different farming systems and agroecological zones globally [6]. The political and economic significance of maize warrants greater attention due to its contribution to food production, employment creation, input supply chains, and the processing industry [7].

Globally, maize is known as the “queen of cereals” due to its exceptional genetic potential compared to other grains [8]. It ranks as the third most important cereal crop worldwide [9]. Maize is cultivated on approximately 196 million hectares globally, yielding 1110 million metric tons of grain annually, with an average productivity of 5.66 tons per hectare [10]. While maize production is abundant across the Americas, Asia, and Europe, consumption is higher in developing countries than in developed regions [1].

Maize was introduced to Africa around 1500 A.D from the Americas and quickly spread across sub-Saharan Africa due to its ease of cultivation and versatility [11], reaching Ethiopia in the 17th century [12]. Despite its importance, maize production faces challenges from environmental and biotic factors [13]. Enhancing food security, nutrition, and economic returns requires increased maize yields, which depend on effective nutrient management to improve productivity and quality under diverse and challenging growing conditions [14]. Optimizing nutrient use is essential for boosting yields and minimizing environmental impacts [15]. The growing demand to meet the food needs of the expanding global population has led to the excessive use of chemical fertilizers, contributing to resource inefficiency and environmental degradation.

Nutrient management strategies for maize vary depending on soil fertility status, climate, farming practices, and other related factors [16]. Effective nutrient management not only improves soil fertility status but also reduces negative environmental outcomes [17]. Maize has high nutrient demands and is particularly sensitive to micronutrient deficiencies [18]. Integrated nutrient management helps provide essential nutrients, addressing deficiencies while curbing the over-application of synthetic fertilizers, a common issue among farmers [19].

Proper nutrient management mitigates environmental risks such as nutrient loss with runoff and soil nutrient degradation, ensuring long-term soil health and productivity [20]. By optimizing the nutrient amount, farmers can improve yields and enhance crop quality and profitability [16]. Efficient nutrient use is critical for agricultural productivity, economic viability, and environmental sustainability [21].

While long-term experimental trials have traditionally been used to evaluate nutrient use efficiency and maize yield, they are resource-intensive and costly. In contrast, crop modeling approaches offer a cost-effective alternative for continuous assessment and informed decision-making [22]. Accurate model calibration using appropriate input data, is essential to ensure reliable predictions of crop growth and yield [23]. Calibration allows models to be applied across various environments and conditions [24]. For instance, the DSSAT model is used to evaluate maize responses to varying sowing dates, cultivars, nitrogen levels, and irrigation practices [25].

The DSSAT was produced by an international group of researchers through the International Benchmark Sites Network initiative to address challenges in agricultural productivity and resources management [26]. As part of the initiative, the CERES-Maize model, part of the DSSAT CSMs, has been widely applied in many nations worldwide [27]. The DSSAT consists of 42 crops and is used to simulate the effects of different practices, such as sowing date, plant population, and cultivar, on plant yield [28]. The DSSAT model integrates weather, crop, management, and soil parameters and determines the combined effects of various strategies on productivity and environmental impacts [29]. The model has been used to assess the impact of climate change on crops [30], optimizes agronomic practices [31], and evaluates cultivar performance and suitability [27,32]. In nutrient management, a 25-year study on sandy loam Alfisol revealed that DSSAT integrates organic and inorganic fertilizers, demonstrating a 20% maize yield increase when NPK is combined with FYM compared to NPK alone, with 93.8% simulation accuracy [33]. Studies using DSSAT have shown that combining chemical fertilizers with manure or straw enhances soil productivity and maize yields by improving soil health and reducing chemical fertilizer dependency [34]. The model has been validated across diverse environments, including tropical, Mediterranean, and temperate regions, accurately representing maize growth and development [35]. However, to ensure reliable predictions and broaden its applicability, extensive datasets are required for proper calibration. The development of DSSAT aimed to facilitate the application of crop system models like the CERES-Maize model in agricultural studies by integrating soil, climate, crop, and management data, enhancing technology dissemination across different regions [36]. Therefore, this review aims to assess the potential of the DSSAT model in promoting effective nutrient management for maize across diverse agroecological zones by simulating and evaluating nutrient use efficiency, yield responses, and environmental impacts. The review seeks to provide valuable insights for decision makers to optimize maize production and promote agricultural productivity.

## 2. Rationality of the Review

The purpose of this review is to provide a comprehensive synthesis of the current knowledge, identify best practices, and highlight future research directions. The findings will be valuable for researchers, agronomists, policymakers, and farmers aiming to enhance maize production through informed nutrient management strategies using the DSSAT model.

## 3. Review Methodology

A systematic review was conducted to investigate maize nutrient management using the DSSAT model. To ensure a comprehensive review, the study applied a structured Preferred Reporting Items for Systematic Reviews and Meta-Analysis (PRISMA) diagram in the review titled “CERES-Maize (DSSAT) Model Applications for Maize Nutrient Management Across Agroecological Zones: A Systematic Review”. The methodology involved searching multiple scholarly databases, including Google Scholar, Pub Med, Web of Science, Agricola, and Scopus. Advanced search strategies such as Boolean operators (AND/OR) and exact phrase searches (“”) were employed alongside simple searches. Keywords and search terms were carefully selected to identify the relevant literature. These included terms such as “DSSAT model”, “Maize”, “Nutrient management”, and “Agro-ecology”, as well as phrases like (“corn or maize or *Zea mays* L. or Indian corn”), “DSSAT”, and (“macronutrients”, “agro-ecology”).

Through extensive database searches using both simple and advanced techniques, 409 articles were identified. Eligibility criteria were applied to ensure the relevance of the sources. The inclusion criteria focused on studies assessing the role of the DSSAT model on maize nutrient management across diverse environmental contexts. Only primary research articles and review papers written in English and published between 2010 and 2024 were considered. Specifically, studies had to apply the DSSAT model to improve maize nutrient management in various agricultural settings. Exclusion criteria included studies unrelated to maize or nutrient management using DSSAT, non-English publications, book chapters, and monographs. Duplicate articles were removed from the dataset, and only studies meeting the predefined inclusion criteria were retained. After screening for relevance, 119 publications (Figure 1) were included in the final review.

## 4. Advantage and Overview of the DSSAT Model

### 4.1. DSSAT’s Development and Features

DSSAT is a robust software suite designed to model growth and yield under various environmental and management scenarios [37]. Developed in the late 1980s and continuously enhanced, it included models for a wide range of crops such as cereals, legumes, root crops, vegetables, fiber, and industrial crops [38]. The models account for factors like soil properties, weather, and management practices, enabling accurate simulations and predictions of crop performance [39,40,41]. DSSAT provides detailed insights into crop responses to different inputs and stresses, supporting decisions in crop management, policymaking, and agricultural research [42]. By combining empirical and mechanistic approaches, DSSAT models plant growth, facilitating the evaluation of climate, soil, and agricultural practices on crop productivity and sustainability [43].

The model effectively simulates agricultural practices by integrating crops, soil carbon and nitrogen, soil moisture, and climate models with crop management options to predict crop growth, yield, and environmental impacts. This versatility has led to its widespread and successful applications globally [44]. The model’s capacity to simulate different management strategies allows users to assess their effects on crop yields and sustainability [45]. Furthermore, the model supports scenario analysis, enabling predictions of how climate change or management practices might impact crop production [46]. This predictive capability is invaluable for strategic planning and informed decision-making in agriculture [43].

### 4.2. Application of DSSAT in Crop Management

#### 4.2.1. DSSAT Tools for Effective Crop Management Practices

##### Optimizing Irrigation Practices with DSSAT

The DSSAT model has been widely used to manage irrigation for wheat, maize, barley, and sunflower crops in Mediterranean regions [47]. The CERES-Rice and CERES-Maize models have also been used to optimize management strategies for rice–maize cropping systems in semi-arid tropical regions [48]. DSSAT enhances the precise simulation of various irrigation strategies, supporting more efficient water use and crop productivity [49]. A novel method for within-season irrigation scheduling was developed in Northwest China by integrating daily yield prediction trends with real-time measurement and historical weather data. This approach allows the DSSAT-CERES-Maize model to adjust forecasts based on actual weather conditions occurring prior to the forecast date [50].

In arid regions, DSSAT identifies the most efficient irrigation intervals to ensure crops receive adequate water during critical growth stages, thereby maximizing water use efficiency and enhancing productivity [51].

The model plays a significant role in agricultural water management by providing detailed data-driven recommendations to improve irrigation practices [52]. DSSAT’s advanced modeling capabilities allow for the simulation of crop growth under various irrigation scenarios by integrating multiple agronomic factors [53]. The accurate calibration of the model and its application for customized irrigation schedules enables farmers to optimize water use, increase crop yields, and promote sustainability [51]. The ability of the model to evaluate different irrigation levels and schedules makes it an invaluable tools for farmers and agricultural planners aiming to enhance productivity while conserving water resources [54].

##### Optimizing Fertilization Strategies

The optimization of nutrient application is essential for balancing crop requirements with fertilizer use, enhancing nutrient use efficiency, and minimizing environmental impacts such as nutrient leaching [55]. Effective fertilization is essential for achieving high crop yields while reducing negative environmental effects [56]. The DSSAT model enables the simulation of the effects of various fertilizer rates and application timings, allowing for the assessment of their impact on crop performance [57]. By facilitating soil fertility management, DSSAT helps farmers optimize fertilizer application, minimize losses, enhance nutrient uptake, and improve organic matter release [58,59]. Fertilization strategies tailored to meet crop nutrient demands reduce waste and increase efficiency. To optimize fertilization practices, DSSAT can simulate extreme nutrient application scenarios, helping identify the best rates and timings for specific crop and soil combinations [32]. This data-driven approach promotes sustainable agriculture by reducing nutrient leaching and mitigating pollution, leading to more efficient and environmentally friendly farming practices [60].

##### Assessing the Impact of Tillage Practices

Process-based simulation models enhance the efficiency of management practices such as tillage, crop diversity, fertilization, crop rotations, drainage, and water use, contributing to sustainable yield and improved environmental quality [26]. Tillage plays a critical role in influencing soil properties, crop growth, and overall agricultural productivity [61,62]. Using DSSAT, researchers can simulate conventional, reduced, and no-till scenarios to evaluate their long-term effects on soil health, moisture retention, nutrient cycling, and crop yields [63]. DSSAT integrates soil, crop, climate, and management data and provides comprehensive insights into how different tillage methods affect maize productivity and sustainability [63]. Through these simulation studies, farmers and researchers can identify the most suitable tillage practices for specific agroecological conditions [64]. Agricultural practices and climate and soil conditions significantly influence soil organic carbon (SOC) dynamics. Adopting practices that minimize soil disturbance and incorporate organic fertilizers can enhance SOC levels, improve soil fertility, and contribute to global greenhouse gas (GHG) mitigation [65].

DSSAT can also be used to simulate no-till farming practices as a strategy to conserve soil water and reduce erosion, thereby boosting crop productivity in dry regions [66,67]. This modeling capability allows farmers to adopt tillage methods that promote soil health, increase moisture retention, and enhance long-term sustainability. Additionally, DSSAT can assess the environmental impacts of different tillage practices, including soil erosion and GHG emissions [68]. The simulation of no-till systems frequently demonstrate lower soil erosion and reduced GHG emissions compared to conventional tillage, reinforcing their role in sustainable farming [68]. For example, a simulation study using a calibrated DSSAT model and projected weather data from 2010 to 2030 to assess the impact of CA and CT practices on productivity under changing climate conditions in Chitedze soils revealed that CA, particularly no-till and crop residue retention, was successful in identifying optimal maize management strategies [67]. No tillage (NT) is increasingly preferred over conventional tillage (CT) for agricultural sustainability, prompting the use of process-based models to simulate their impacts because long-term, site-specific field experiments are resource-intensive [65].

##### Evaluating Crop Rotation Systems

The DSSAT model has been applied at the regional level to evaluate the effectiveness of crop rotation systems [69]. Implementing diverse crop rotation provides numerous benefits, including reduced production risks and uncertainties, enhanced soil and ecological sustainability, and increased income diversification [70]. DSSAT allows for the simulation and assessment of various crop rotation schemes, offering insights into their effects on soil quality and crop yields [69]. Crop rotation sequences, particularly those alternating legumes with cereals, play a significant role in sustainable farming by improving soil structure, reducing bulk density, enhancing aggregation, and increasing porosity and saturated hydraulic conductivity [71].

By modeling the outcomes of different rotation sequences, DSSAT helps farmers identify optimal rotations that enhance soil nutrient balance and disrupt pest and disease cycles [72]. For example, DSSAT simulations have demonstrated that incorporating leguminous crops into cereal-based rotations can significantly increase soil nitrogen levels, fostering more sustainable and efficient agricultural systems [73]. This evidence empowers farmers to design crop rotation schedules that maximize yields while preserving long-term soil health [74].

##### Evaluating Crop Varieties

DSSAT facilitates the evaluation of new crop varieties by simulating their performance under diverse environmental and management conditions [75,76]. This capability allows researchers and agronomists to assess the impact of various nutrient management strategies across different climatic conditions for different varieties [75]. By performing simulation studies, informed decisions can be made regarding which varieties to prioritize in breeding programs [77,78]. A simulation study used the DSSAT model in Tanzania’s Morogoro, Kilosa, Kongwa, Kiteto, and Kilindi districts to identify the best suited varieties (Pioneer Phb 3253, Situka, Staha, and TMV1) in each district [78]. Similarly, DSSAT was used to evaluate the performance of five maize varieties (BH-540, BH-546, BH-547, Shala, and Shone) across four locations (Shamana, Bilate, Hawassa, and Dilla) in Ethiopia using 30 years of weather data and location-specific management practices [76]. The DSSAT-CSM is also employed to evaluates the impact of climate change and develop adaptation strategies for maize in Ethiopia. This work focuses on three varieties (BH-660, BH-540, Melkasa-1), planting dates, nitrogen rates, and irrigation levels to optimize yields and improve resilience [79].

#### 4.2.2. Supporting Sustainable Agriculture

Sustainable agriculture has evolved into a multifaceted endeavor encompassing various biological, chemical, and physical processes associated with soil, water, climate, and crop management [80]. Crop simulation models play a crucial role in optimizing management strategies to achieve sustainable outcomes, such as predicting climate impacts, managing resources, and improving agricultural practices [55]. The DSSAT model exemplifies this by advancing sustainable agricultural production and enhancing crop management techniques [28,81].

The effectiveness of DSSAT underscores the importance of decision-support tools in preserving agricultural resources and promoting long-term sustainability [82]. Agricultural management practices can be tailored based on key factors such as soil composition, variety choice, irrigation methods, fertilizer application, and climatic conditions [83]. By simulating various scenarios, DSSAT highlights the benefits of sustainable farming practices including improved soil quality, reduced erosion, and lower GHG emissions [84].

##### Reducing Environmental Impact

The DSSAT plays a crucial role in optimizing agricultural resource management by simulating crop growth, soil conditions, and climatic interactions [85]. Through calibration and evaluation using field data, DSSAT accurately predicts crop productivity under various management practices and climate conditions [39]. This capability is essential for promoting sustainable and environmentally friendly farming by providing precise models that guide tailored interventions that meet crop demands while protecting natural resources [86].

DSSAT also facilitates the evaluation and adoption of conservation practices aimed at improving soil health and biodiversity [87]. These practices enhance soil structure, increase organic matter, and support beneficial soil organisms, contributing to a more resilient agricultural ecosystem [88]. Additionally, DSSAT helps reduce the carbon footprints of farming operations, promoting sustainable agriculture aligned with conservation goals and long-term environmental stewardship [84].

##### Enhancing Food Security

Food availability, a fundamental component of food security, is predicted and or forecast through crop model simulations that predict grain yield and biomass outputs [89]. Crop modeling enhances global food and nutrition security by improving management practices and supporting the development of resilient crop varieties [90]. DSSAT plays a significant role in this process by fine-tuning predictive models, enabling farmers to increase harvests and mitigate losses caused by adverse weather or pests [55]. The application of crop simulation models in multi-cropping systems is a key strategy for achieving sustainable food security and efficient resource management [87]. DSSAT has been successfully employed to assess the vulnerability and risk associated with food production systems in response to climate change. This allows agriculturalists to develop necessary adaptation strategies, fostering more resilient cropping systems that ensure stable global crop production [87]. For instance, agroecological and multi-cropping approaches, which enhance nutrient and water efficiency while promoting ecosystem services, can be simulated through DSSAT to support global food security efforts [87].

##### DSSAT and Maize Nutrient Management Improvements

DSSAT is an effective modeling tool widely used to optimize maize nutrient management [33]. By simulating crop growth and development under various environmental and management scenarios, DSSAT helps identify optimal nutrient application rates and timings [33,45]. This is especially important for maize, a crop with high nutrient demands. DSSAT’s predictive capabilities enable farmers to foresee nutrient deficiencies and adjust fertilization plans, leading to increased yields and reduced input costs [55].

The model integrates weather, soil, management, and crop data to provide a comprehensive analysis of nutrient dynamics in maize production [91]. Its ability to simulate diverse scenarios allows researchers and agronomists to evaluate the effects of different nutrient management strategies across various climate conditions [92]. For example, DSSAT can simulate the effects of different nitrogen application rates on maize yield, offering insights into practices that improve nitrogen use efficiency [58,93]. This not only enhances crop yields but also helps addresses environmental concerns like nitrate leaching and nitrous oxide emissions [94]. Researchers have applied the CERES-Maize model to explore nitrogen and phosphorus management strategies, optimizing maize yields across multiple agroecological zones [95].

These studies provide critical insights into boosting production under varying climate conditions. DSSAT continues to be a valuable tool for optimizing maize nutrient management globally, offering practical recommendations to improve yield, sustainability, and resilience. The following table (Table 1) summarizes key case studies, highlighting the diverse applications and impacts of DSSAT in maize nutrient management studies across different agricultural strategies.

## 5. CERES-Maize (DSSAT) Model Performance in Different Environmental Settings

DSSAT models effectively simulate crop growth and yield across diverse environmental conditions, although their performance can vary depending on factors such as water consumption and productivity [100]. The performance and productivity of maize cropping systems differ significantly across agroecological zones [101]. For example, maize cultivation in Benin requires substantial nutrient inputs, particularly nitrogen and phosphorus, to support soil fertility and achieve optimal yields [96]. Conversely, in Kenya, limited and erratic rainfall, poor soil moisture conservation, and degraded soils with low nutrient inputs result in consistently low maize yields [102].

Therefore, the application of DSSAT plays a vital role in optimizing maize nutrient management across diverse environmental conditions [37]. By simulating maize growth under different management practices and environmental conditions, the model assists to identify the best nutrient application strategies, enhancing both productivity and sustainability [33]. One of DSSAT’s key advantages is its ability to integrate pertinent biophysical factors that affect crop performance and simulate their impacts [33]. Maize grown in a temperate climate may have different nutrient requirements than maize grown in tropical or arid regions [103]. DSSAT can accommodate these environmental variations, enabling precise fertilizer adjustments tailored to specific environmental conditions [104].

Through DSSAT simulations, farmers can predict nutrient needs and adapt their management practices to ensure maize receives the necessary nutrient under various environmental stresses [32]. Beyond individual farm management, DSSAT’s applications extend to agricultural policy formulations and research [28]. Researchers globally use the model to evaluate long-term impacts of nutrient management strategies on soil health and maize yields at field, regional, and global scales, providing valuable insights for shaping sustainable agricultural policies [57]. Additionally, DSSAT supports the development of region-specific nutrient management recommendations, enhancing extension services and ensuring farmers receive tailored, context-specific guidance [105]. Overall, DSSAT bridges the gap between scientific research and practical agricultural applications, fostering improved nutrient management for maize cultivation in diverse environmental settings [106].

Crop models such as DSSAT are essential for assessing the potential impacts of climate change on farming systems and quantifying the role of nutrient management to adapt to the impact, aiding in the development of proactive strategies to adapt to and mitigate these effects [27]. Table 2 below summarizes key case studies, illustrating the diverse applications of DSSAT under different environmental conditions.

## 6. Challenges and Limitations in DSSAT Applications

Crop simulation models, including DSSAT, are valuable tools for predicting yield and water use, but they come with limitations and uncertainties that must be addressed to improve efficiency and foster collaboration with the software industry [26,109]. DSSAT, for instance, lacks models for several crops and factors that affect crop performance and yield [28]. Additionally, challenges arise from the absence of a standardized minimum dataset (MDS) definition for crop modeling inputs, as well as inconsistencies in input and output file formats across different models [110]. Overcoming these limitations is essential to ensure reliable agricultural solutions [30].

Despite the wide variety of crops globally, DSSAT currently supports only 42 crops [111]. Furthermore, the model may not fully capture all interactions within the plant–soil–atmosphere continuum [112]. The application of DSSAT in fruit and vegetable crop cultivation is limited, particularly in crop rotations, sequence analysis, and economic evaluations—key areas for promoting environmental and economic sustainability [100].

Some biotic and abiotic factors influencing plant growth and production may not be fully accounted for within the model [89]. Crop simulation modeling inherently involves uncertainty due to complex interactions among soil, climate, crop and management practices [75]. Single-model simulations cannot quantify this uncertainty, and studies have demonstrated significant variability in their results [113]. Addressing these limitations is crucial to improving the model’s applicability and reliability across diverse agricultural contexts [114]. While a single model may perform well in specific regions and agroecological conditions, multi-model comparison approaches are recommended to enhance yield estimation accuracy and uncertainty analysis. Ensemble modeling, incorporating multiple crop models, offers a more robust solution for mitigating uncertainties [115].

For instance, in Ethiopia, three crop models—APSIM–Maize, CERES–Maize, and Aqua-Crop—were calibrated to simulate maize growth under local conditions [116]. While APSIM-Maize and CERES-Maize accurately predicted flowering and maturity dates, CERES-Maize overestimated maturity date for late varieties in high-elevation agroecology. The AquaCrop model more effectively simulated maize canopy cover with an nRMSE less than 10.8%. An ensemble approach combining these models improved simulation accuracy and reduced uncertainty [116].

In Chilanga, Zambia, APSIM-Maize and CERES-Maize models were calibrated and evaluated, yielding RMSE values for days to anthesis (APSIM 1.91 days; CERES-Maize 2.89 days) and maturity (APISM 3.35 days; CERES-Maize 4.13 days), with errors of less than 5%. Both models simulated grain yield with low RMSE (APSIM-Maize 1.38 t/ha and CERES-Maize 0.84 t/ha), grain number per square meter, and soil water content across layers 1–8 (RMSE ≤ 20%), demonstrating reliability in rainfed conditions with overall RMSE values below 30% [117].

In Brazil, three crop simulation models—AEZ-FAO, CERES-Maize, and APSIM-Maize—performed well, with MAEs from 727 to 1376 kg ha^−1^ and R^2^ values between 0.49 and 0.79. An ensemble of CERES-Maize and APSIM-Maize further improved accuracy, yielding an MAE of 799 kg ha^−1^ and R^2^ of 0.79, d of 0.94, and C of 0.84, significantly reducing uncertainty in maize yield predictions [118]. Data assimilation techniques, which addresses model input uncertainties by incorporating remote sensing observations of crop characteristics, have shown promise in improving model calibration and enhancing prediction accuracy [119].

## 7. Discussion

The CERES-Maize (DSSAT) model plays a critical role in nutrient management to enhance maize productivity by providing a platform to evaluate different management strategies and predict their performance under diverse environmental conditions. Its extensive testing across tropical, Mediterranean, and temperate regions demonstrate its ability to accurately simulate maize growth and development. This adaptability highlights the model’s reliability and value in supporting nutrient management across diverse agroecological zones. By integrating various components—such as crop system models, soil carbon models, nitrogen models, daily water balance models, and crop/land management options, DSSAT accurately simulates crop growth, yields, and environmental impacts. This comprehensive approach enables researchers and farmers to explore and refine management strategies to improve productivity while maintaining sustainability. Addressing model performance challenges has driven the development of ensemble models like CERES-Maize and APSIM-Maize. These ensemble approaches enhance maize yield predictions, reduce uncertainties, and improve the overall reliability of model outputs. A detailed review of DSSAT’s application in maize nutrient management offers valuable insights into current practices, highlights areas for improvement, and identifies directions for future research. This review serves as a critical resource for researchers, agronomists, policymakers, and farmers seeking to optimize maize production through effective nutrient management strategies. Efficient nutrient management is essential for maximizing maize yield quality, addressing factors that limit productivity and ensuring long-term agricultural sustainability.

## 8. Conclusions

The DSSAT model plays a crucial role in optimizing maize nutrient management by simulating crop performance under diverse environmental and management conditions. Its ability to simulate the impact of irrigation, fertilization, crop rotations, and variety evaluations supports sustainable agricultural practices and enhances productivity. By integrating crop–soil–weather interactions, DSSAT provides a cost-effective approach to inform decision-making and improve resource efficiency.

While DSSAT has demonstrated success in identifying the best management options that improve maize yields and sustainability, challenges such as data requirements, calibration complexities, and model uncertainties remain. The application of multiple models and data assimilation from remote sensing has shown promise in improving accuracy and reducing uncertainty.

Maize’s significance in global food security highlights the importance of continued research and innovation in nutrient management. As a research and decision support tool, DSSAT provides valuable insights for identifying optimal strategies to enhance maize production. Its contributions assist researchers, agronomists, policymakers, and farmers in improving yields and building resilience against environmental and agronomic challenges.

## Figures and Tables

**Figure 1 plants-14-00661-f001:**
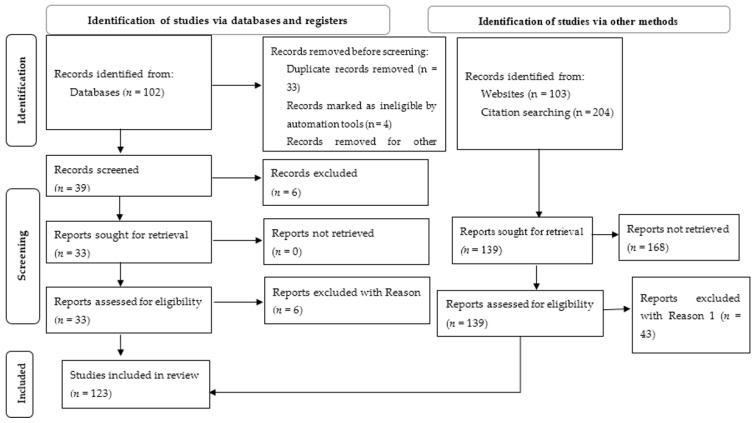
PRISMA flow diagram depicting the study selection process. This diagram illustrates the identification, screening, eligibility assessment, and inclusion of studies for the systematic review. It includes records identified from databases, registers, websites, and citation searches, as well as the reasons for exclusion at each stage.

**Table 1 plants-14-00661-t001:** Diverse applications and impacts of DSSAT in maize nutrient management under various agricultural strategies.

Location	Agroecology	Focus	Methodology	Findings	Reference
Benin’s Sudano-Guinean zones	Agroecology includes a rainy season (mid-April to October), Ferric and Plintic Luvisols in the Sudano-Guinean zone, Acrisols in the “Terre de barre”, and cropping systems of maize, cassava, yam, cotton, and cashew.	Fertilizer recommendations for maize.	A two-year study (2011–2012) validated DSSAT model predictions.	Strong correlation between observed and predicted yields (R^2^ = 80–94%). Proper N-P-K rates, along with crop residue management and organic manure, improve maize productivity. The DSSAT model is effective for optimizing fertilizer recommendations in Benin.	[96]
Nigeria’s savannas	Agroecology characterized by low-fertility loamy to silt loam soils, rainfall ranging from 795 to 1611 mm, amidst climate variability.	Evaluating nitrogen and phosphorus fertilization for maize adaptation to climate change.	Long-term simulations to observe variability in maize yields due to climate uncertainty.	N and P fertilizers improved maize yields, with nitrogen as a key limiting factor for maize production.	[97]
Huang-Huai-Hai Plain, China	A temperate climate with average solar radiation of 2857.59 MJ/m^2^, effective accumulated temperature of 1624.72 °C, and annual rainfall of 291.97 mm.	Optimizing nitrogen application and planting density for maize cultivation.	Adjusting nitrogen inputs and planting density through simulations.	Achieving up to 73% of maize’s potential yield with optimized nitrogen and planting density.	[58]
Trans Nzoia County, Kenya	A bimodal rainfall pattern (900–1700 mm annually), a mean temperature of 18 °C, and fertile soils including Humic and Rhodic Nitisols and Humic Ferralsols supporting diverse crop production.	Evaluation of the DSSAT–CERES-Maize model to assess the maize yield response of two common cultivars under various agricultural strategies, such as sowing dates, nitrogen fertilization, and water management.	Calibration and evaluation of the DSSAT–CERES-Maize model long-term yields (1984–2021) and characterize production under various weather regimes.	Application of 100 kg N ha^−1^ and early sowing spanning mid-February to mid-March has the greatest potential for enhancing maize yields in the study region.	[98]
Lusaka, Zambia	Characterized by a subtropical climate with warm, wet summers and cool, dry winters, receiving annual rainfall of 800–1000 mm.	Evaluate the CERES-Maize model in simulating planting date, nitrogen, and yield. Determine model accuracy in predicting phenology, biomass, and grain yield.	The performance of the CERES-Maize model was assessed through a field trial conducted in Agro-ecological Region II of Zambia, known for its subtropical climate.	The model showcased a remarkable level of precision in estimating phenology, biomass production, grain yield, and soil moisture content.	[99]
New Delhi, Indian Trans-Gangetic Plains Zone (Agro Climatic Zone-VI)	Characterized by a semi-arid climate with seasonal rainfall, specific soil properties influencing crop growth.	To calibrate the CERES-Maize model (DSSAT v 4.8) for simulating maize growth, yield, and nitrogen dynamics in a long-term CA system, investigating temperature, soil nitrate, ammonia concentration, and the impact of tillage and nitrogen management on yields.	CERES-Maize (DSSAT) model was used to simulate growth and nitrogen dynamics under diverse nitrogen management options within a CA-based maize–wheat system. Model performance was evaluated using prediction error, RMSE, and regression analysis.	DSSAT model simulated maize phenological stages accurately.	[57]
Gongzhuling, Jilin Province, Northeast China	Defined by a cool climate with seasonal rainfall, fertile black soil affected by declining organic carbon, and long-term fertilization practices combining chemical and organic inputs.	Simulating spring maize yields, impact of long-term fertilization on BSP, predicting long-term yield trends.	Used the DSSAT model to simulate yields under different fertilization treatments, analyzed basic soil productivity (BSP).	Demonstrated strong correlation with actual data (RMSE ranging from 9.70 to 13.46%), provided insights into the impact of long-term fertilization on BSP, and showcased the model’s reliability in simulating crop growth under varying conditions. Long-term yield trends were accurately predicted.	[34]

**Table 2 plants-14-00661-t002:** Diverse applications and impacts of DSSAT under different environmental conditions.

Location	Agroecology	Focus	Methodology	Findings	References
Pak Chong, northeastern Thailand	Features a tropical climate where solar radiation, CO_2_ levels, and temperature variations significantly influence maize growth and yield.	Analyzing the effects of climate change on maize growth and yield.	Utilized the DSSAT software to simulate the effects of climate change on maize growth and yield in Thailand.	Simulation results reveal that solar radiation and CO_2_ levels significantly influence maize yield, and temperature changes affect anthesis and maturity days, thus impacting maize production.	[107]
Northern Ghana	A temperate continental semi-arid region with 450 mm annual rainfall, high evaporation (1563.3 mm), neutral Loess soils (pH 7.4) low in organic matter.	Maize yield gaps due to soil productivity and erratic rainfall, climate variability impact on maize yield.	CERES-Maize model performance was evaluated using RMSE and Willmott’s d-index; evaluated model was applied to predict maize response. Variance analysis (ANOVA) was conducted to assess yield differences across various management systems.	Maize yield increased with nitrogen application, while soil productivity and rainfall variability significantly influenced the results. Climate variability continues to affect crop production across sub-Saharan Africa.	[106]
Ethiopia	Semi-arid “Woina Dega” zones with clay loam and loam soils, moderate rainfall (649 mm at Mekelle, 675 mm at Maychew), and seasonal temperatures of 13–26 °C and 10.7–22 °C, respectively, semi-arid “Woina Dega” zones with clay loam and loam soils, moderate rainfall (649 mm at Mekelle, 675 mm at Maychew), and seasonal temperatures of 13–26 °C and 10.7–22 °C, respectively.	Employ the model to evaluate climate change impacts on maize yield. Assess maize adaptation strategies under future climate.	A simulation study using the DSSAT-CSM CERES-Maize model evaluated the impact of climate change on maize productivity in various maize varieties, planting dates, N fertilizer rates, and water levels.	Maize yield changes ranged from −13% to +13% relative to the baseline, driven by an average temperature increase of 2.7–3.0 °C by mid-century. Yield reductions were predicted for the BH-540 and Melkasa-1 varieties.	[79]
Chilanga, Zambia	Experiences semi-arid conditions with 930 mm seasonal rainfall, average temperatures of 21.8 °C (max 28.3 °C, min 15.4 °C), and Ustic Paleustalf soils characterized by neutral-to-slightly-alkaline pH (6.2–6.8), low organic carbon (0.35–0.80%), and low nitrogen levels (0.031–0.061%).	Calibrate and validate CERES-Maize models to simulate phenology, mLAI, soil water content, aboveground biomass, and grain yield under rainfed and irrigated conditions.	Data from a split–split plot design field experiment used to evaluate the CERES-Maize models.	The model accurately simulates phenology, growth, yield, and soil water content.	[108]

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
