# Peer review of "CERES-Maize (DSSAT) Model Applications for Maize Nutrient Management Across Agroecological Zones: A Systematic Review"

_plants, 2025, doi:10.3390/plants14050661_

Round 1

Reviewer 1 Report

Comments and Suggestions for Authors

The manuscript entitled "CERES-Maize (DSSAT) Model Applications for Maize Nutrient Management Across Agroecological Zones: A Systematic Review" contains interesting results regarding CERES-Maize (DSSAT).
This is a review paper on an important and current topic for science and agricultural practice.

The paper should be improved before publication.
I have included detailed comments in the original PDF text.

General comments:

Add the Latin name of maize to the keywords. It will make it easier to search databases

Write the Latin name of the species in italics (line 26)

line 27 correct the word "maie"

Provide (write) the latest available DSSAT model

lines 127 and 128 correct the spelling of two words (comments in the text)

123 literature items were used for the study. I quote: "After screening for relevance, 123 publications (Figure 1) were included in the final review. List them all in the bibliography.

line 208 remove the dot

You can combine chapter 7 and 8 into one general conclusion. I leave this for the authors to consider

Correctly write the authors' contribution to the manuscript

Add the Abbreviations used in the manuscript, e.g.:
DSSAT - ?
CERES-maize - ?
APSIM-Maize
SOC
GHG
NT
CT

e.t.c

Improve the list of references. Especially items 8, 9, 58, 63, 66, 77,

I hope my comments will help the authors improve the text of the manuscript. Best regards

Author Response

Dear reviewer, I tried to address all the comments given and attached below. 

Reviewer 2 Report

Comments and Suggestions for Authors

Authors have produced a literature-based compilation of existing knowledge about models for predicting crop development and yield as function of various environmental data (climate, weather, soil quality) and implemented agronomical measures (soil cultivation and fertilization, irrigation, cultivar). Based on the presented findings, such models can help farmers in decision-making process due to (more or less reliable) prediction of effects that agronomic measures can bring to crop yield. Uncertainty of model predictions stems from impossibility to run the calibration field trials with all available cultivars, at all known soils and sites, at all climatic and microclimatic conditions and at all possible climatic aberrations. However, for improving the reliability of such models, there is needed a fine calibration of models in long-term field trials at many different sites and soils, with a spectrum of various agronomy practices. The work is concentrated on nutrient management for maize and can inform the potential users about benefits from using such models as decision-support tool. Also, it can help merchants with agricultural commodities to predict the production volume for shaping the pricing policy.

There are some mistakes that need to be corrected:

Line 26. in brackets “Zea mays L.“ has to be written in italic font.

Line 28. “mos“ change into “most“.

Line 29. “palys“ change into “plays“.

Line 36. “ints“ change into “its“.

Line 38. “exceptionalgenetic“ separate into “exceptional genetic“.

Line 41. after “grain” add “annually”.

Line 53. After “inefficiency” add “and”.

Line 59. Separate “acommon” into “a” and “common”.

Line 76. “CERES-maizde” change into “CERES-maize”.

Line 77. “appliedin” into “applied in”.

Line 87. “shownthat” into “shown that”.

this review aims to assess the potential of the DSSAT model in promoting effective nutrient management for maize across diverse agroecological zones

line 97. “zoes” into “zones”.

Line 127. “non-inclish” into “non-english”.

Line 272. “improves” into “improved”.

Line 366. After “come” insert “with”, “addressin” change into “addressed”.

Author Response

(The authors gave the same response as above.)
